# Oxygen–Ion Conductivity, Dielectric Properties and Spectroscopic Characterization of "Stuffed" $Tm_2(Ti_{2-x}Tm_x)O_{7-x/2}$ ($x$ = 0, 0.1, 0.18, 0.28, 0.74) Pyrochlores

**Nikolay Gorshkov** [1,2], **Egor Baldin** [1], **Dmitry Stolbov** [3], **Viktor Rassulov** [4], **Olga Karyagina** [5] **and Anna Shlyakhtina** [1,*]

1   N.N. Semenov Federal Research Center for Chemical Physics, Russian Academy of Sciences, 4 Kosygina Street, Building 1, 119991 Moscow, Russia; gorshkov.sstu@gmail.com (N.G.); baldin.ed16@physics.msu.ru (E.B.)

2   Department of Chemistry and Technology of Materials, Yuri Gagarin State Technical University of Saratov, 77 Polytecnicheskaya Street, 410054 Saratov, Russia

3   Department of Chemistry, Lomonosov Moscow State University, 1-3 Leninskiye Gory, GSP-1, 119991 Moscow, Russia

4   N.M. Fedorovsky All-Russian Scientific Research Institute of Mineral Raw Materials, 31 Staromonetny ln, 119017 Moscow, Russia

5   Emanuel Institute of Biochemical Physics RAS, Russian Academy of Sciences, 4 Kosygina Street, 119334 Moscow, Russia

*   Correspondence: annash@chph.ras.ru or annashl@inbox.ru

**Abstract:** $Tm_2(Ti_{2-x}Tm_x)O_{7-x/2}$ ($x$ = 0, 0.1, 0.18, 0.28, 0.74) solid electrolytes have been investigated as potential electrolyte materials for solid oxygen fuel cells (SOFCs), operating in the medium temperature range (600–700 °C). The design of new oxygen-conducting materials is of importance for their possible utilization in the solid oxide fuel cells. The oxygen–ion conductivity of the $Tm_2(Ti_{2-x}Tm_x)O_{7-x/2}$ ($x$ = 0, 0.1, 0.18, 0.28, 0.74) "stuffed" pyrochlores ceramics was investigated by electrochemical impedance spectroscopy (two-probe AC) in dry and wet air. The synthesis of precursors via co-precipitation and the precipitate decomposition temperature have been shown to be of key importance for obtaining dense and highly conductive ceramics. At ~770 °C, the highest total conductivity, ~$3.16 \times 10^{-3}$ S/cm, is offered by $Tm_2Ti_2O_7$. The conductivity of the fluorite-like solid solution $Tm_2(Ti_{2-x}Tm_x)O_{7-x/2}$ ($x$ = 0.74) is an order of magnitude lower. However, for the first time a proton contribution of ~$5 \times 10^{-5}$ S/cm at 600 °C has been found in $Tm_2(Ti_{2-x}Tm_x)O_{7-x/2}$ ($x$ = 0.74) fluorite. Until now, compositions with proton conductivity were not known for the intermediate and heavy rare earth titanates $Ln_2(Ti_{2-x}Ln_x)O_{7-x/2}$ ($Ln$ = Ho − Lu) systems. The use of X-ray diffraction (structural analysis with Rietveld refinement), optical spectroscopy and dielectric permittivity data allowed us to follow structural disordering in the solid solution series with increasing thulium oxide content. High and low cooling rates have been shown to have different effects on the properties of the ceramics. Slow cooling initiates' growth of fluorite nanodomains in a pyrochlore matrix. The fabrication of such nanostructured dense composites is a promising direction in the synthesis of highly conductive solid electrolytes for SOFCs. We assume that high-temperature firing of nanophase precursors helps to obtain lightly doped "stuffed" pyrochlores, which also provide the high oxygen–ion conductivity.

**Keywords:** $Tm_2Ti_2O_7$; "stuffed" pyrochlores; co-precipitation method; oxygen–ion conductivity

## 1. Introduction

$Ln_2M_2O_7$ ($Ln$ = La–Lu, M = Ti, Zr, Hf) pyrochlores are currently thought to be a viable alternative to yttria- and scandia-stabilized $ZrO_2$ in its main applications, such as electrolytes for solid-oxide fuel cells [1–5], membranes for pure oxygen production [6] and heat-resistant materials for thermal barrier coatings [7,8]. In addition, there are some

other active applications: e.g., catalysis [9,10], ferroelectrics [11] and luminescence [5,12,13]. Considerable attention is currently being paid to the possibility of utilizing rare-earth (RE) zirconates and hafnates with the pyrochlore structure as materials for the immobilization of actinide-rich nuclear wastes [14]. This diversity of properties can be understood in terms of specific features of the phase diagrams of $Ln_2O_3$-$MO_2$ (M = Ti, Zr, Hf) systems, in which compounds with the pyrochlore structure have broad isomorphism ranges, occasionally up to ~30 mol%. This allows single-phase materials to be readily synthesized in wide ranges of temperatures and $Ln_2O_3$ concentrations [15]. The reason for the flexibility of the systems in question is that they typically have pyrochlore–fluorite (order–disorder) transitions [16,17], which allows one to follow changes in properties due to a gradual transition from one crystal structure to another. In recent years, considerable attention has been paid to so-called "stuffed" pyrochlores, in which M cations are partially replaced by the lanthanide [18–22]. As shown in spectroscopic studies of broad isomorphism ranges, such materials are inhomogeneous in a number of cases and their short-range structure contains fluorite nanodomains, which can only be detected using spectroscopic techniques [5,20,21]. Nanoinhomogeneity can help improve the conductivity and luminescence of high-temperature ceramics [5]. Mullens et al. [20] prepared a $Tm_2(Ti_{2-x}Tm_x)O_{7-x/2}$ ($x = 0$–0.67) series by solid-state reaction and measured the oxygen–ion conductivity of these mixed oxides. In a study of pyrochlore $Tm_2Ti_2O_7$ prepared via co-precipitation followed by freeze drying of the precursor [23], a three orders of magnitude higher conductivity ($2 \times 10^{-3}$ S/cm at 740 °C) was found in comparison with a sample having the same nominal composition and prepared by solid-state reaction ($3 \times 10^{-6}$ S/cm at 800 °C) [20].

Pandit et al. [24] studied the ac and dc conductivity and dielectric properties of a flux-grown $Tm_2Ti_2O_7$ single crystal in the range 27–727 °C. The crystal was grown from a 7.2 $Tm_2O_3$ + 3.2 $TiO_2$ + 20.0 $MoO_3$ + 28.0 $PbF_2$ + 28.0 PbO + 2.0 $PbO_2$ high-temperature solution during slow cooling from 1270 to 800 °C [25]. $Tm_2Ti_2O_7$ was shown to be a *p*-type semiconductor with a band gap of 2.96 eV and mixed ionic–electronic conductivity in the range 27–377 °C. In the range 377–727 °C, electron mobility and concentration increased, whereas the ionic conductivity of the titanate became negligible. The temperature dependence of its permittivity had an anomaly near 377 °C, which was attributed to a change in conduction mechanism.

Using neutron diffraction, Ross et al. [26] compared the structure of a float-zone $Yb_2Ti_2O_7$ single crystal ground into powder (Tm and Yb are neighbors in the lanthanide series) and that of ceramics with the same composition, $Yb_2Ti_2O_7$. Rietveld refinement of neutron powder diffraction data for these materials showed that the ground crystal could be best described as a "stuffed" pyrochlore, $Yb_2(Ti_{2-x}Yb_x)O_{7-x/2}$ with $x = 0.046$, even though the starting material had the stoichiometric composition. Unfortunately, the oxygen–ion conductivity of the single crystal was not reported [26]. Nominally stoichiometric $Yb_2Ti_2O_7$ ceramics prepared using co-precipitation [27] are known to have the highest conductivity ($1 \times 10^{-2}$ S/cm at 800 °C) among all of the RE titanates reported to have oxygen–ion conductivity. At the same time, in the $Yb_2(Ti_{2-x}Yb_x)O_{7-x/2}$ series prepared by solid-state reaction, the highest conductivity was offered by the $Yb_2(Ti_{2-x}Yb_x)O_{7-x/2}$ ($x = 0.07$) solid solution, but it was rather low: ~$5 \times 10^{-5}$ S/cm at 800 °C [21].

Note that conductivity maxima reported for $Ln_2(Ti_{2-x}Ln_x)O_{7-x/2}$ "stuffed" pyrochlore systems with various lanthanides do not always coincide. In the thulium system, $Tm_2(Ti_{2-x}Tm_x)O_{7-x/2}$ ($x = 0$–0.67), the 800 °C conductivity has a maximum, $1 \times 10^{-5}$ S/cm, at $x = 0.268$ [20]. The lutetium system, $Lu_2(Ti_{2-x}Lu_x)O_{7-x/2}$, has a conductivity maximum at $x = 0.1$: $6 \times 10^{-3}$ S/cm at 800 °C [28]. The yttrium system, $Y_2(Ti_{2-x}Y_x)O_{7-x/2}$, has a maximum at $x = 0.29$: ~$1 \times 10^{-3}$ S/cm at 800 °C [29]. In $Ho_2(Ti_{2-x}Ho_x)O_{7-x/2}$, the maximum is located at $x = 0.48$: ~$5 \times 10^{-3}$ S/cm at 800 °C [19]. One possible reason for this is that, at a given stoichiometry, the synthesis method and temperature, as well as the ionic radius ratio R$Ln$/RTi, influence the degree of structural disorder in the pyrochlores, which certainly influences their oxygen–ion conductivity [30]. As mentioned above, the $Yb_2(Ti_{2-x}Yb_x)O_{7-x/2}$ series prepared by solid-state reaction has the highest conductivity,

~5 × 10$^{-5}$ at 800 °C, at $x$ = 0.07 [21], whereas in the case of co-precipitation, the highest 800 °C conductivity, 1 × 10$^{-2}$ S/cm, is offered by nominally stoichiometric pyrochlore $Tm_2Ti_2O_7$ [27].

Thus, there are still significant discrepancies in the conductivity data reported for oxygen–ion conducting RE titanates prepared by different techniques, and the origin of the striking difference in properties between the $Tm_2Ti_2O_7$ single crystal and ceramics with the nominal composition $Tm_2Ti_2O_7$ remains unclear. In a study of oxygen–ion conducting ceramics and a single crystal with the composition $La_{10}W_2O_{21}$ and a fluorite-related structure [31], the ionic conductivity of the crystal was found to considerably exceed that of the ceramics. The conductivity of a hexagonal $La_2W_{1+x}O_{6+3x}$ ($x$~0.22) single crystal was reported to be lower than that of $La_{18}W_{10}O_{57}$ ceramics, having a similar composition and the same structure [32], but above 600 °C they had the same conduction mechanism: oxygen–ion transport. At the same time, below 600 °C the temperature behavior of conductivity for the single crystal changed sharply, and the activation energy for conduction increased to 2.17 eV [32].

In this paper, we report the properties of a series of $Tm_2(Ti_{2-x}Tm_x)O_{7-x/2}$ ($x$ = 0, 0.1, 0.18, 0.28, 0.74) prepared via co-precipitation followed by high-temperature firing. The materials were characterized by impedance spectroscopy (oxygen–ion conductivity and dielectric properties) during both heating and cooling. Optical spectroscopy was used to follow the variation in the degree of structural order with Tm content on the Ti site. We make assumptions that account for the large spread in the conductivity data for $Tm_2(Ti_{2-x}Tm_x)O_{7-x/2}$ materials prepared by different methods and the difference in properties between the $Tm_2Ti_2O_7$ single crystal and ceramics with the same nominal composition.

## 2. Materials and Methods

Precursors for the synthesis of $Tm_2(Ti_{2-x}Tm_x)O_{7-x/2}$ ($x$ = 0, 0.1, 0.18, 0.28, 0.74) ceramics were prepared by co-precipitation. The starting materials used were $Tm_2O_3$ (99.99%, TuO-1, purity standard TU 48-4-182-74, Moscow, Russia) and $TiCl_4$ (OSCh 12-3, TU-6-09-2118-77, Moscow, Russia). $Tm_2O_3$ was dissolved in hydrochloric acid, and the titer of the solution was determined gravimetrically. The titanium-containing starting reagent used was $TiCl_4$ dissolved in concentrated hydrochloric acid. The titer of this solution was also determined gravimetrically. Co-precipitation from the thulium and titanium solutions in hydrochloric acid was performed at pH 11 using aqueous ammonia as a precipitant. The resultant precipitates were centrifuged and repeatedly washed with water. The precipitates were dried in a drying oven at 105 °C for 24 h and then decomposed at 650 °C for 2 h, following which the precursors were pressed at 140 MPa and fired at 1600 °C for 4 h. After the firing, the samples were furnace-cooled (2.2 °C/min). In addition, $Tm_2(Ti_{2-x}Tm_x)O_{7-x/2}$ ($x$ = 0.28, 0.74) ceramics were prepared using slow cooling: the material was reheated to 1600 °C, slowly cooled at 3 °C/h to 1300 °C and then furnace-cooled.

Absorption spectra of the materials were calculated from their diffuse reflectance spectra, measured on a TerraSpec 4 Hi-Res spectrophotometer (Malvern Panalytical, Malvern, UK) from 350 to 2500 nm (28,000 to 4000 cm$^{-1}$) at a spectral resolution of 3 nm in the range 350 to 1000 nm and 6 nm in the rest of the spectrum. The response function of the spectrophotometer was taken into account at regular time intervals using a Labsphere certified Spectralon reflectance standard (Labsphere, Inc., North Sutton, USA). Absorption bands were fitted with a set of Gaussians using Origin version 8 (OriginLab Corp., Northampton, MA, USA). The estimate of the approximation accuracy is the coefficient of determinism $R^2$.

Samples for impedance measurements had the form of disks 7.7 mm in diameter and ~2.5 mm in thickness. Porous platinum coatings on their faces were used as electrodes. Oxygen–ion conductivity was determined by impedance measurements in dry air (for all samples) and wet air (for fluorite-like $Tm_2(Ti_{2-x}Tm_x)O_{7-x/2}$ ($x$ = 0.74) only) in the range 255–771 °C (heating and cooling in about 50 °C steps with 2-h holding at each temperature)

using a Novocontrol Alpha AN (Novocontrol Technologies GmbH and Co. KG, Montabaur, Germany) impedance meter. The frequency range was 0.1 Hz to 1 MHz, and the applied ac voltage amplitude was varied from 50 to 500 mV.

XRD patterns were collected at room temperature with a Rigaku Smartlab SE X-ray diffractometer (Cu $K_\alpha$ radiation, $\lambda$ = 1.5418 Å, Bragg-reflection geometry, 40 kV, 50 mA) in continuous mode. The 2θ range was 10° to 70°, scan step 0.1°, scan rate 5°/min. Rietveld refinement determination was carried out using the SmartLab Studio II version 4.4 software. In some cases, X-ray diffraction patterns for the powder samples were collected at room temperature on a DRON-3M automatic diffractometer (Cu $K_\alpha$ radiation, $\lambda$ = 1.5418 Å, Bragg-reflection geometry, 35 kV, 28 mA) in the 2θ range of 10° to 75° (scan step of 0.1° τ = 3 s).

The microstructure of the ceramic samples was examined using scanning electron microscopy (SEM) on a JEOL JSM-6390LA (JEOL, Tokyo, Japan).

## 3. Results and Discussion

### 3.1. Structure of the Solid Solutions Studied by XRD and Optical Spectroscopy

Figure 1a presents Rietveld refinement results for the nominally stoichiometric pyrochlore phase. According to XRD results, the $Tm_2(Ti_{2-x}Tm_x)O_{7-x/2}$ ($x$ = 0, 0.1, 0.18, 0.28) samples had a pyrochlore long-range order, whereas the highly substituted $Tm_2(Ti_{2-x}Tm_x)O_{7-x/2}$ material ($x$ = 0.74) had the fluorite structure.

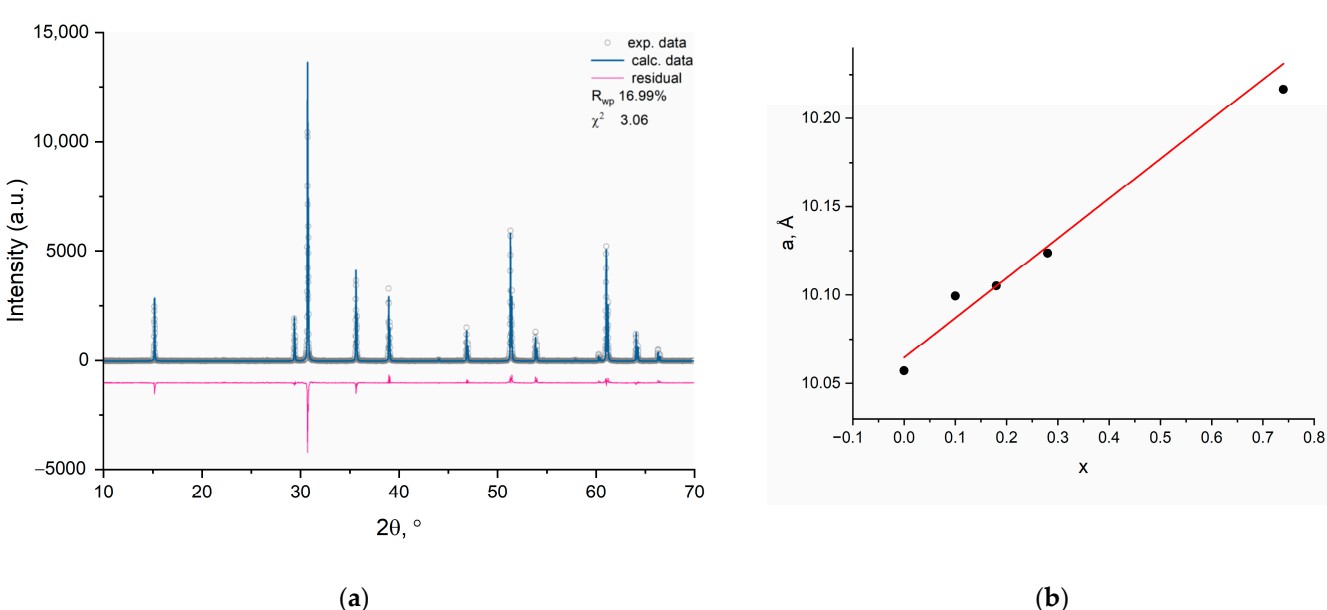

(**a**)　　　　　　　　　　　　　　　　　　　　(**b**)

**Figure 1.** (**a**) Rietveld refinement results for stoichiometric pyrochlore $Tm_2Ti_2O_7$ and (**b**) composition dependence of the unit-cell parameter for the $Tm_2(Ti_{2-x}Tm_x)O_{7-x/2}$ ($x$ = 0, 0.1, 0.18, 0.28, 0.74) materials prepared by firing at 1600 °C for 4 h.

Figure 1b shows the composition dependence of the unit-cell parameter for the $Tm_2(Ti_{2-x}Tm_x)O_{7-x/2}$ ($x$ = 0, 0.1, 0.18, 0.28, 0.74) ceramics prepared at 1600 °C (4 h). It is worth noting that the unit-cell parameter increases with Tm content on the Ti site ($RTi^{4+}_{CN=6}$ = 0.605 Å; $RTm^{3+}_{CN=6}$ = 0.88 Å). No impurity phases were detected in the ceramics.

According to recent work [5,20], the broad isomorphism range of the pyrochlore structure observed in some titanate [20] and zirconate [5] systems does not exist in short-range order, and the materials contain nanodomains of related phases. Given this, in addition to conventional furnace cooling, we carried out controlled slow furnace cooling of the $Tm_2(Ti_{2-x}Tm_x)O_{7-x/2}$ ($x$ = 0, 0.28) sample, similar in nominal composition to the material having the highest conductivity in the thulium system: $Tm_2(Ti_{2-x}Tm_x)O_{7-x/2}$

($x = 0.268$) prepared by solid-state reaction [20]. Using Rietveld refinement of structural parameters after rapid and slow cooling, we calculated the structure of the resultant ceramics in two models (Table 1):

1.  $Tm_2(Ti_{2-x}Tm_x)O_{7-x/2}$ ($x = 0.28$) as a single-phase titanate pyrochlore and
2.  $Tm_2(Ti_{2-x}Tm_x)O_{7-x/2}$ ($x = 0.28$) as a mixture of two (pyrochlore + fluorite) phases.

**Table 1.** Calculation of the structure of the rapidly and slowly cooled $Tm_2(Ti_{2-x}Tm_x)O_{7-x/2}$ ($x = 0.28$) and $Tm_2(Ti_{2-x}Tm_x)O_{7-x/2}$ ($x = 0.74$) ceramics with Rietveld refinement in three models: (1) pure pyrochlore, (2) pyrochlore + fluorite impurities and (3) fluorite.

| Composition, Cooling Mode | Pyrochlore (P) | Pyrochlore + Fluorite (P + F) | Fluorite (F) |
|---|---|---|---|
| $Tm_2(Ti_{2-x}Tm_x)O_{7-x/2}$ ($x = 0.28$) Fast cooling | Rwp = 12.62% Rp = 10.10% S = 1.19 $\chi^2$ = 1.41 a = 10.1241(9) Å | Rwp = 12.68% Rp = 10.12% S = 1.18 $\chi^2$ = 1.39 P = 99.32 wt% F = 0.68 wt% a = 10.1261(8) Å | - |
| $Tm_2(Ti_{2-x}Tm_x)O_{7-x/2}$ ($x = 0.28$) Slow cooling | Rwp = 12.00% Rp = 9.40% S = 1.15 $\chi^2$ = 1.33 a = 10.1226(2) Å | Rwp = 11.46% Rp = 9.18% S = 1.09 $\chi^2$ = 1.20 P = 98.61 wt% F = 1.39 wt% a = 10.1229(7) Å | - |
| $Tm_2(Ti_{2-x}Tm_x)O_{7-x/2}$ ($x = 0.74$) Fast cooling | Rwp = 14.27% Rp = 10.36% S = 1.33 $\chi^2$ = 1.77 a = 10.2158(11) Å | Rwp = 10.47% Rp = 8.41% S = 0.98 $\chi^2$ = 0.95 P = 6.5 wt% F = 93.5 wt% a = 10.215 (4) Å | Rwp = 10.58% Rp = 8.46% S = 0.99 $\chi^2$ = 0.97 a = 10.2153(8) Å |
| $Tm_2(Ti_{2-x}Tm_x)O_{7-x/2}$ ($x = 0.74$) Slow cooling | Rwp = 13.78% Rp = 10.54% S = 2.29 $\chi^2$ = 5.27 a = 10.23738(47) Å | Rwp = 13.06% Rp = 9.51% S = 2.15 $\chi^2$ = 4.64 P = 13 wt% F = 87 wt% a = 10.23574(1) Å | Rwp = 13.19% Rp = 9.52% S = 2.21 $\chi^2$ = 4.90 a = 10.2494(6) Å |

Note that, in the case of the slowly cooled sample, the two-phase model ensures better results (all *R*-factors are smaller), whereas in the case of the rapidly cooled pyrochlore phase, none of the models is preferable. It is reasonable to assume that the size of the fluorite nanodomains in the pyrochlore matrix slightly increased during slow cooling.

A fluorite-like solid solution $Tm_2(Ti_{2-x}Tm_x)O_{7-x/2}$ ($x = 0.74$), which may contain pyrochlore nanodomains (~2 nm) in the short-range order, as previously found for $Yb_2(Ti_{2-x}Yb_x)O_{7-x/2}$ ($x = (x = 0.67)$ by the TEM method [33], was cooled at different rates. In this case, the two-phase model (fluorite + pyrochlore) was also the best fit, regardless of the cooling rate.

According to the Rietveld refinement, the fluorite phase content reaches 87 and 93.5 wt.% for the slowly and rapidly cooled sample $Tm_2(Ti_{2-x}Tm_x)O_{7-x/2}$ ($x = 0.74$), respectively (Table 1).

Figure 2 shows the absorption spectra for samples of the $Tm_2(Ti_{2-x}Tm_x)O_{7-x/2}$ ($x = 0$, 0.1, 0.18, 0.28, 0.74) series of solid solutions obtained by conventional furnace cooling. The absorption spectra of the ceramics show broad, slightly structured bands due to transitions from the $Tm^{3+}$ $^3H_6$ ground state to the $^3F_4$ (5840 cm$^{-1}$), $^3H_5$ (8460), $^3H_4$ (12700), $^3F_3$ +

$^1F_2$ (14650) and $^1G_4$ (21370) excited multiplets. With increasing Tm content on the Ti site, the structural flexibility of the $Tm_2(Ti_{2-x}Tm_x)O_{7-x/2}$ system leads to a pyrochlore-to-fluorite phase transition, accompanied by marked disordering, which shows up in optical absorption spectra of the ceramics.

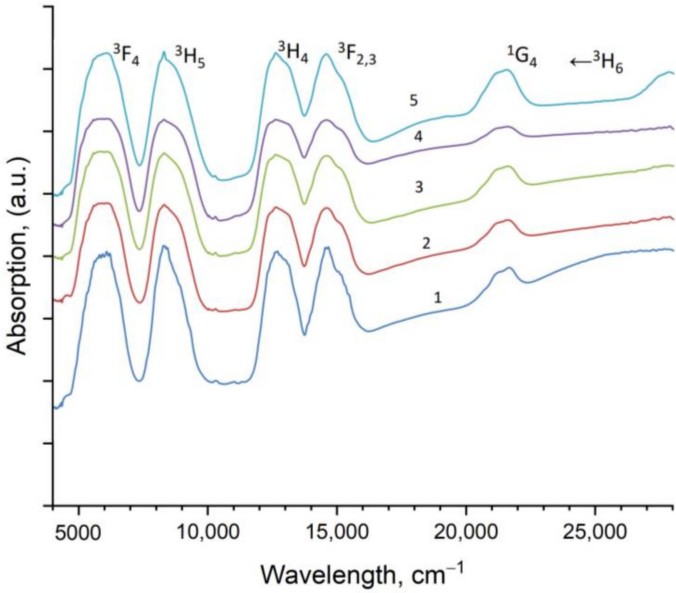

**Figure 2.** Absorption spectra of the $Tm_2(Ti_{2-x}Tm_x)O_{7-x/2}$ "stuffed" solid solutions: $x$ = (1) 0, (2) 0.1, (3) 0.18, (4) 0.28, (5) 0.74.

The absorption bands of $Tm^{3+}$ become inhomogeneously broadened. To follow structural changes, we used the $^3H_6 \rightarrow {}^3H_5$ transition, which can be thought of as supersensitive. Energy-level transitions meeting the selection rules $\Delta J \leq 2$, $\Delta L \leq 2$ and $\Delta S = 0$ [34] (where $\Delta J$, $\Delta L$ and $\Delta S$ are changes in the optical transition of the spin S orbital L and the total angular momentum of the momentum J, written according to the $^{2S+1}L_J$ scheme) are thought to be supersensitive to the RE environment. The results obtained by decomposing the $^3H_6 \rightarrow {}^3H_5$ band into two Gaussians are presented in Table 2. The center frequency of the band peaking near 8000 cm$^{-1}$ gradually decreases as the Tm content of the $Tm_2(Ti_{2-x}Tm_x)O_{7-x/2}$ solid solutions increases to $x$ = 0.28 and then rises sharply as the percentage of the fluorite phase in the ceramics increases. The full width at half maximum (FWHM) of this band gradually increases with Tm content. The position of the band peaking near 8800 cm$^{-1}$ varies little with Tm content, whereas its FWHM gradually increases.

**Table 2.** Decomposition of the $^3H_6 \rightarrow {}^3H_5$ band into Gaussians.

| Curve No. in Figure 2 | Percent $Tm_2O_3$, % | $x$ in $Tm_2(Ti_{2-x}Tm_x)O_{7-x/2}$ | $R^2$ | First Band | | | Second Band | | |
|---|---|---|---|---|---|---|---|---|---|
| | | | | Center | Width | Height | Center | Width | Height |
| 1 | 33.3 | 0 | 0.999 | 8146 | 613 | 0.412 | 8808 | 861 | 0.374 |
| 2 | 35 | 0.1 | 0.999 | 8057 | 609 | 0.288 | 8767 | 925 | 0.359 |
| 3 | 37.5 | 0.18 | 0.999 | 8026 | 632 | 0.254 | 8773 | 1067 | 0.362 |
| 4 | 40 | 0.28 | 0.998 | 7977 | 674 | 0.261 | 8811 | 1141 | 0.381 |
| 5 | 52 | 0.74 | 0.999 | 8054 | 714 | 0.290 | 8797 | 1134 | 0.402 |

The above data demonstrate that the absorption spectra of the solid solutions are sensitive to changes in their structure. In particular, the spectra of the $Tm_2(Ti_{2-x}Tm_x)O_{7-x/2}$ ($x$ = 0, 0.1, 0.18, 0.28) materials indicate that, with increasing Tm content, their pyrochlore structure undergoes disordering, whereas the spectrum of the $Tm_2(Ti_{2-x}Tm_x)O_{7-x/2}$ ($x$ = 0.74) ceramic corresponds to a fluorite derivative structure, with an additional absorption band at ~28,000 cm$^{-1}$. The present results confirm that the supersensitive transitions in

absorption spectra of $Tm^{3+}$ are structure-sensitive; i.e., they are influenced by structural disordering with increasing Tm content, but we failed to detect fluorite nanodomains in the $Tm_2(Ti_{2-x}Tm_x)O_{7-x/2}$ ($x$ = 0, 0.1, 0.18) pyrochlore solid solutions by this spectroscopic technique.

It is of interest to note that a defect fluorite structure and the structure of amorphous defect fluorite-like phases resulting from crystallization of pyrochlores were recently described in terms of short-range order using a weberite-based structural model (sp. gr. *Ccmm*), which ensured the best fit [16,17,35]. Most likely, the 28,000 $cm^{-1}$ band observed in the spectrum of the fluorite phase is assignable to nanodomains with the weberite structure (sp. gr. *Imma*), which is also a fluorite derivative. In many systems where pyrochlores crystallize, there exist weberites as well. The stability of the B cations in the pyrochlore structure is important for its stabilization, e.g., in $Ca_{2-x}Cd_xSb_2O_7$ and $Ca_{2-x}Cd_xSb_2O_6F$ substituted calcium antimonates [36].

### 3.2. Microstructure of the $Tm_2(Ti_{2-x}Tm_x)O_{7-x/2}$ ($x$ = 0, 0.1, 0.18, 0.28, 0.74) Ceramics

A study of the microstructure of thermally etched stuffed pyrochlore $Tm_2(Ti_{2-x}Tm_x)O_{7-x/2}$ ($x$ = 0, 0.1, 0.18) ceramics showed that the ceramics are composed of grains of 1–12 μm in size and that there are practically no pores in them (Figure 3a–c). The $Tm_2(Ti_{2-x}Tm_x)O_{7-x/2}$ ($x$ = 0.1) ceramic is more homogeneous. The particle size distribution is between 1 and 7 μm (Figure 3c). The grains are in the form of polyhedrons with 4–6 faces. Small crystallites up to 1 μm in size were observed on the ceramic surface. It is most likely that these are small particles of the micro-sized crystalline precursor that were not incorporated into the grain during crystallization, forming a fine-grained fraction. Figure 3d shows a fracture surface of the $Tm_2(Ti_{2-x}Tm_x)O_{7-x/2}$ ($x$ = 0.28) ceramic. It can be seen that there is virtually no closed porosity in this ceramic.

The ceramic with the highest content of $Tm_2O_3$–$Tm_2(Ti_{2-x}Tm_x)O_{7-x/2}$ ($x$ = 0.74) has a different microstructure (Figure 3d,f). Its grains are elongated in one direction, and there is no open porosity. It is evident that an unusual striated microstructure is preserved inside the grains (Figure 3f). The resulting ceramics have a density close to 100%.

### 3.3. Conductivity of the Solid Solutions Studied by Impedance Spectroscopy

Figure 4a–d shows impedance spectra of $Tm_2(Ti_{2-x}Tm_x)O_{7-x/2}$ ($x$ = 0, 0.1, 0.18, 0.28) measured during cooling in dry air. The impedance plots each have the form of three semicircles and can be interpreted using an equivalent circuit (Figure 4a–d, inset) comprising three series connected elements: (RbCPEb), (RgbCPEgb) and (ReCPEe), where $R_b$ is the bulk resistance; $R_{gb}$ is the grain boundary resistance; $R_e$ is the electrode–ceramic interface resistance; and CPEb, CPEgb and CPEe are constant phase elements. The impedance spectra of the $Tm_2(Ti_{2-x}Tm_x)O_{7-x/2}$ ($x$ = 0, 0.1, 0.18, 0.28) pyrochlores are well resolved (Figure 4a–d). We can distinguish bulk, grain boundary and electrode contributions. At the same time, with increasing Tm content, the grain boundary component disappears, and the spectrum of the fluorite-related $Tm_2(Ti_{2-x}Tm_x)O_{7-x/2}$ ($x$ = 0.74) (Figure 4e) consists of only two semicircles, attributable to the bulk and electrode components of the impedance.

Figure 5 shows temperature dependences of total and bulk conductivity, and Table 3 presents activation energies for conduction. The conductivity of the $Tm_2(Ti_{2-x}Tm_x)O_{7-x/2}$ ($x$ = 0, 0.1, 0.18, 0.28) ceramics obeys the Arrhenius law in a wide temperature range, from 255 to 771 °C. It is also worth noting that, almost throughout the temperature range studied, the highest conductivity ($3.16 \cdot 10^{-3}$ S/cm at 770 °C) is offered by the nominally stoichiometric pyrochlore phase $Tm_2Ti_2O_7$ (Table 3), like in previously studied ytterbium titanate-based ceramics [26]. Recently, a maximum oxygen–ion conductivity of ~$9.4 \times 10^{-2}$ S/cm at 800 °C has been achieved on 5 mol% $Bi_2O_3$ doped ScSZ ($Sc_2O_3$ doped $ZrO_2$) [37]. However, the introduction of $Bi_2O_3$ oxide has its negative side associated with its easy reduction. With increasing Tm content, the activation energy for bulk conduction in the $Tm_2(Ti_{2-x}Tm_x)O_{7-x/2}$ ($x$ = 0, 0.1, 0.18) solid solutions gradually rises from 0.946 to 1.071 eV (Table 3).

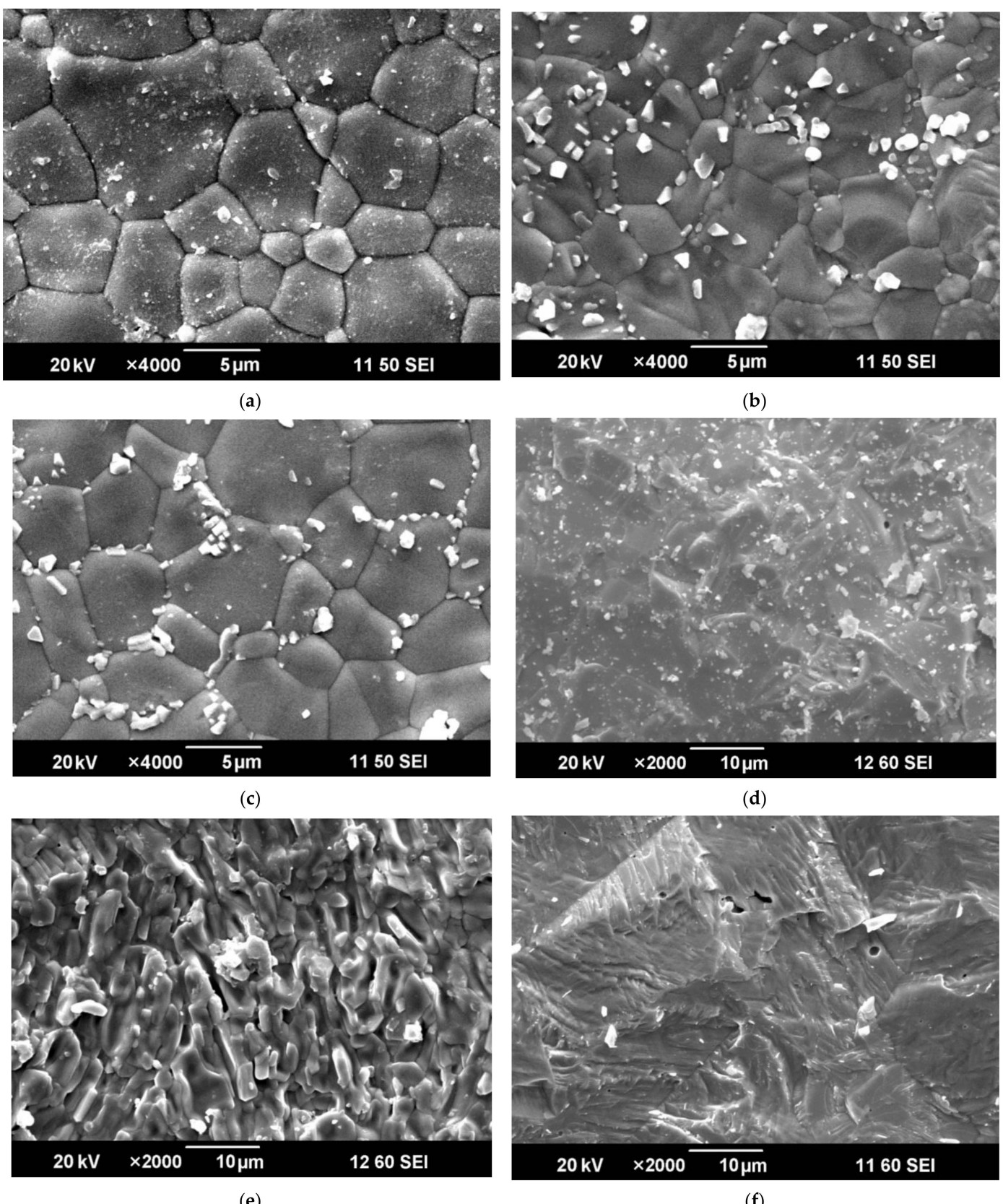

**Figure 3.** Microstructure of (**a**–**c**,**e**) an outer surface and (**d**,**f**) of the fracture surface of the Tm$_2$(Ti$_{2-x}$Tm$_x$)O$_{7-x/2}$ ceramics: $x$ = (**a**) 0, (**b**) 0.1, (**c**) 0.18, (**d**) 0.28, (**e**,**f**) 0.74.

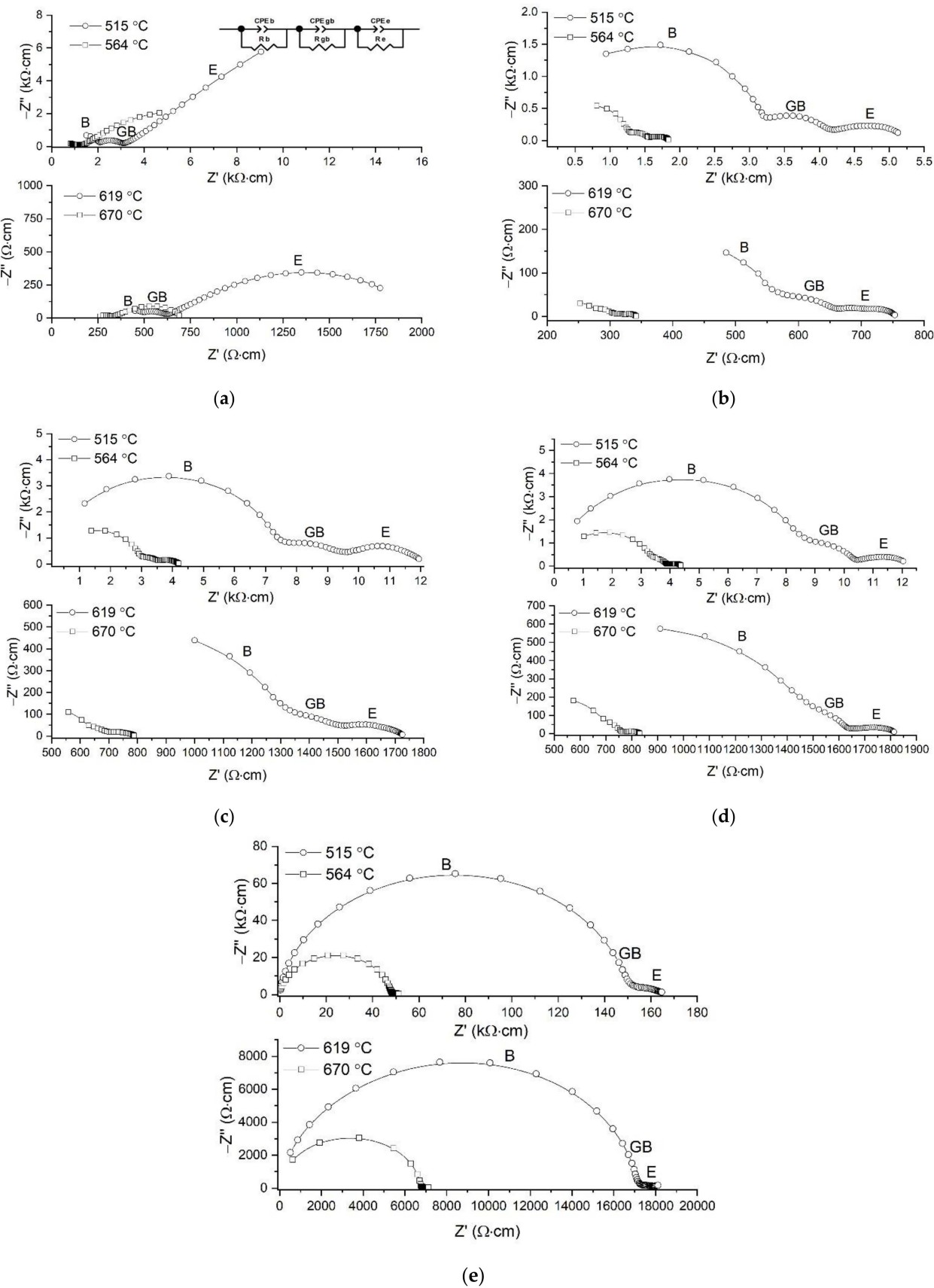

**Figure 4.** Impedance spectra of the $Tm_2(Ti_{2-x}Tm_x)O_{7-x/2}$ "stuffed" solid solutions: $x =$ (**a**) 0, (**b**) 0.1, (**c**) 0.18, (**d**) 0.28, (**e**) 0.74. B—Impedance of grain bulk, GB—Impedance of grain boundaries, E—Impedance of electrode–ceramic interface.

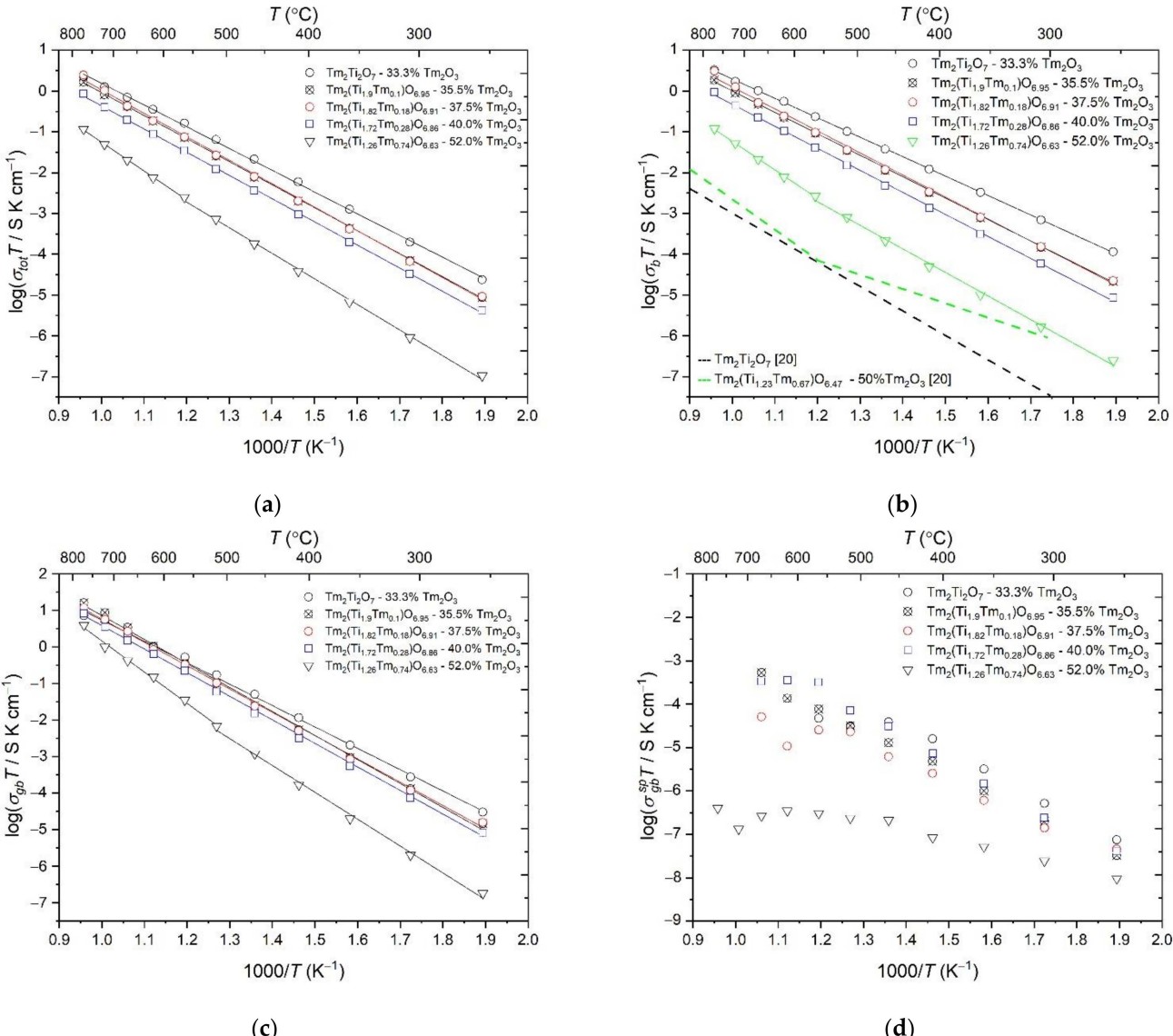

(a) (b)

(c) (d)

**Figure 5.** Arrhenius plots of the (**a**) total, (**b**) bulk conductivity for the nominally stoichiometric pyrochlore phases prepared in this study using coprecipitation and by Mullens et al. [20] and for fluorite-related solid solutions: $Tm_2(Ti_{2-x}Tm_x)O_{7-x/2}$ ($x = 0.74$) (coprecipitation, this work) and $Tm_2(Ti_{2-x}Tm_x)O_{7-x/2}$ ($x = 0.67$) (solid-state reaction [20]), and (**c**) ($\sigma_{gb}^{app}$) apparent conductivity calculated from impedance spectra and (**d**) specific grain-boundary conductivity ($\sigma_{gb}^{sp} = (C_1/C_2) \times \sigma_{gb}^{app}$) for the $Tm_2(Ti_{2-x}Tm_x)O_{7-x/2}$ "stuffed" solid solutions.

Figure 5c,d show two sets of calculated grain boundary conductivity data. Figure 5c presents apparent grain-boundary conductivity ($\sigma_{gb}^{app}$) data obtained directly from the impedance spectra, whereas Figure 5d presents specific grain-boundary conductivity $\sigma_{gb}^{sp} = (C_1/C_2) \times \sigma_{gb}^{app}$ obtained with the $C_1/C_2$ coefficient, where $C_1$ and $C_2$ are the capacitances of the grain bulk and grain-boundary components attained by the deconvolution of the impedance spectra using the following assumptions in the brick layer model [38–42]. This model relies on the following assumptions: (1) $\sigma_{bulk} > \sigma_{gb}^{sp}$, (2) g (grain boundary thickness) << G (diameter of the grains) and (3) $\varepsilon_{gb}$ (the permittivity of the grain-boundary phase) $\sim \varepsilon_{bulk}$ (the permittivity of the grain interior phase). Note that it may require some revision based on new experimental data because there is solid evidence that, in some cases, grain boundary conductivity considerably exceeds bulk conductivity [43,44].

**Table 3.** Characteristics of the conductivity of materials under study.

| Samples No | Composition, %, Cooling Mode | Composition, $Tm_2(Ti_{2-x}Tm_x)O_{7-x/2}$, Cooling Mode | Bulk Conductivity at 771 °C (S/cm) | | Apparent Activation Energy (Ea) of Bulk Conductivity in Dry and Wet Air, eV | Apparent Activation Energy (Ea) of Total Conductivity in Dry and Wet Air, eV |
|---|---|---|---|---|---|---|
| 1 | 33.3% $Tm_2O_3$ + 66.7% $TiO_2$, fast cooling | $Tm_2(Ti_{2-x}Tm_x)O_{7-x/2}$ (x = 0), fast cooling | $3.16 \cdot 10^{-3}$ | | 0.946 | 1.058 |
| 2 | 35.5% $Tm_2O_3$ + 64.5% $TiO_2$, fast cooling | $Tm_2(Ti_{2-x}Tm_x)O_{7-x/2}$ (x = 0.1), fast cooling | $1.78 \cdot 10^{-3}$ | | 1.043 | 1.121 |
| 3 | 37.5% $Tm_2O_3$ + 62.5% $TiO_2$, fast cooling | $Tm_2(Ti_{2-x}Tm_x)O_{7-x/2}$ (x = 0.18), fast cooling | $2.96 \cdot 10^{-3}$ | | 1.071 | 1.141 |
| 4 | 40.0% $Tm_2O_3$ + 60.0% $TiO_2$, fast cooling | $Tm_2(Ti_{2-x}Tm_x)O_{7-x/2}$ (x = 0.28), fast cooling | $0.91 \cdot 10^{-3}$ | | 1.067 | 1.123 |
| 4S | 40.0% $Tm_2O_3$ + 60.0% $TiO_2$, slow cooling | $Tm_2(Ti_{2-x}Tm_x)O_{7-x/2}$ (x = 0.28), slow cooling | $1.36 \cdot 10^{-3}$ | | 1.062 | 1.096 |
| 5 | 52.0% $Tm_2O_3$ + 48.0% $TiO_2$, fast cooling | $Tm_2(Ti_{2-x}Tm_x)O_{7-x/2}$ (x = 0.74), fast cooling | $0.11 \cdot 10^{-3}$ | 771–515 °C | 1.380 | 1.339 |
| | | | | 515–255 °C | 1.119 | 1.222 |
| 5S | 52.0% $Tm_2O_3$ + 48.0% $TiO_2$, slow cooling | Dry air | $0.10 \cdot 10^{-3}$ | 771–515 °C | 1.381 | 1.404 |
| | | $Tm_2(Ti_{2-x}Tm_x)O_{7-x/2}$ (x = 0.74), slow cooling | | 515–255 °C | 1.142 | 1.186 |
| | | Wet air | $0.13 \cdot 10^{-3}$ | 771–515 °C | 1.228 | 1.276 |
| | | | | 515–255 °C | 1.091 | 1.092 |

Figure 5b shows Arrhenius plots of bulk conductivity for the thulium series in comparison with data reported by Mullens et al. [20]. It is seen that, at the nominal composition $Tm_2Ti_2O_7$, the difference in conductivity between the samples prepared in this study using co-precipitation and those prepared by solid-state reaction [20] is up to three orders of magnitude. The plot for fluorite-related $Tm_2(Ti_{2-x}Tm_x)O_{7-x/2}$ (x = 0.74) has a small break at 550 °C, like in the case of $Tm_2(Ti_{2-x}Tm_x)O_{7-x/2}$ (x = 0.67) prepared by solid-state reaction [20].

Above 550 °C, the fluorite-like ceramics prepared by the methods in question differ little in activation energy for conduction, whereas between 300 and 500 °C there is a considerable difference. This means that the choice of the synthesis method plays an important role in the preparation of pyrochlores. It seems likely that, in the synthesis of these phases, good mixing of their constituent cations should be ensured. Powders and, accordingly, ceramics prepared by conventional solid-state reaction are typically slightly inhomogeneous [45].

The conductivity of the $Tm_2(Ti_{2-x}Tm_x)O_{7-x/2}$ (x = 0, 0.1, 0.18, 0.28) pyrochlores measured during heating differed insignificantly from that obtained during cooling. At the same time, in the case of the fluorite-related $Tm_2(Ti_{2-x}Tm_x)O_{7-x/2}$ (x = 0.74) material, there was a marked difference in both total and bulk conductivity (Figure 6). The difference in temperature behavior is due to the inhomogeneity of the composite ceramics, which contain domains of one phase embedded in the matrix of another phase. In the above case, the fluorite phase was a matrix. Clearly, in the case of the samples prepared by the wet chemical method, which are more uniform in composition and microstructure, the highest oxygen–ion conductivity is offered by the composite pyrochlores, whereas the composite fluorites have much lower conductivity. Unlike Mullens et al. [20], we detected no high conductivity in the case of the fluorite-related solid solutions. Thus, the matrix of such a high-temperature composite should have a basic pyrochlore structure to obtain high conductivity. Conductivity depends on the change in occupancy of the 48f, 8a and 8b sites in the $Tm_2(Ti_{2-x}Tm_x)O_{7-x/2}$ system and is consistent with previous results [18,46,47]. The fluorite-based composite has substantially lower conductivity, in conflict with previously

reported data [20]. Note that, in the $Yb_2(Ti_{2-x}Yb_x)O_{7-x/2}$ series [21], synthesized by the same group as the thulium series [20], also using solid-state reaction, the conductivity of pyrochlore $Yb_2Ti_2O_7$ ceramics was substantially higher than that of fluorite-related $Yb_2(Ti_{2-x}Yb_x)O_{7-x/2}$ ($x = 0.67$) ceramics, in full accord with previously reported conductivity data for such "stuffed" titanate series [19,27–29], and in both cases the conductivity was extremely low.

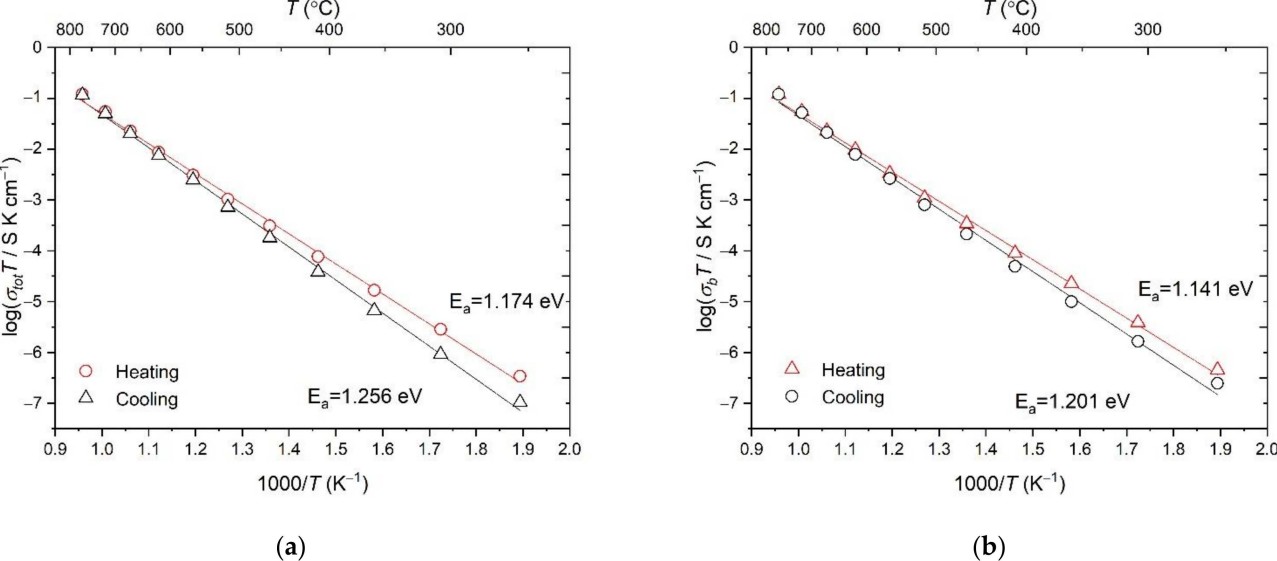

(**a**)                                                                           (**b**)

**Figure 6.** Arrhenius plots of (**a**) total and (**b**) bulk conductivity measured during heating and cooling for fluorite-related $Tm_2(Ti_{2-x}Tm_x)O_{7-x/2}$ ($x = 0.74$).

Figure 7 presents bulk conductivity data for two $Tm_2(Ti_{2-x}Tm_x)O_{7-x/2}$ ($x = 0.28$) samples, after rapid and slow cooling. The conductivity of the slowly cooled sample is seen to slightly exceed that of the rapidly cooled sample. It is reasonable to assume growth of fluorite nanodomains in the pyrochlore matrix to an optimal size, which ensures higher conductivity of such an ultradense "composite."

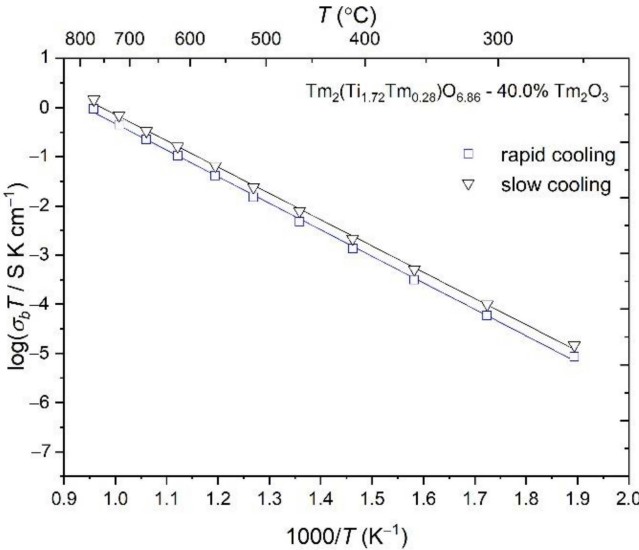

**Figure 7.** Arrhenius plots of bulk conductivity for the rapidly and slowly cooled $Tm_2(Ti_{2-x}Tm_x)O_{7-x/2}$ ($x = 0.28$) samples.

Comparison of the bulk conductivity data obtained for the rapidly and slowly cooled $Tm_2(Ti_{2-x}Tm_x)O_{7-x/2}$ ($x = 0.28$) samples during heating and cooling (Figure 8) indicates

that these samples have slightly different Arrhenius plots of conductivity. The conductivity of the slowly cooled sample exhibits different types of behavior during heating and cooling, which is typical of fluorite-related ceramics (Figure 6). Note that, being more homogeneous, the ceramics prepared via co-precipitation behave as a uniform system. They have the smallest possible scatter in grain size and consist of more homogeneous grains, so they cool uniformly. It seems likely that ceramics prepared by solid-state reaction have a considerable scatter in grain size, and such systems typically cool nonuniformly.

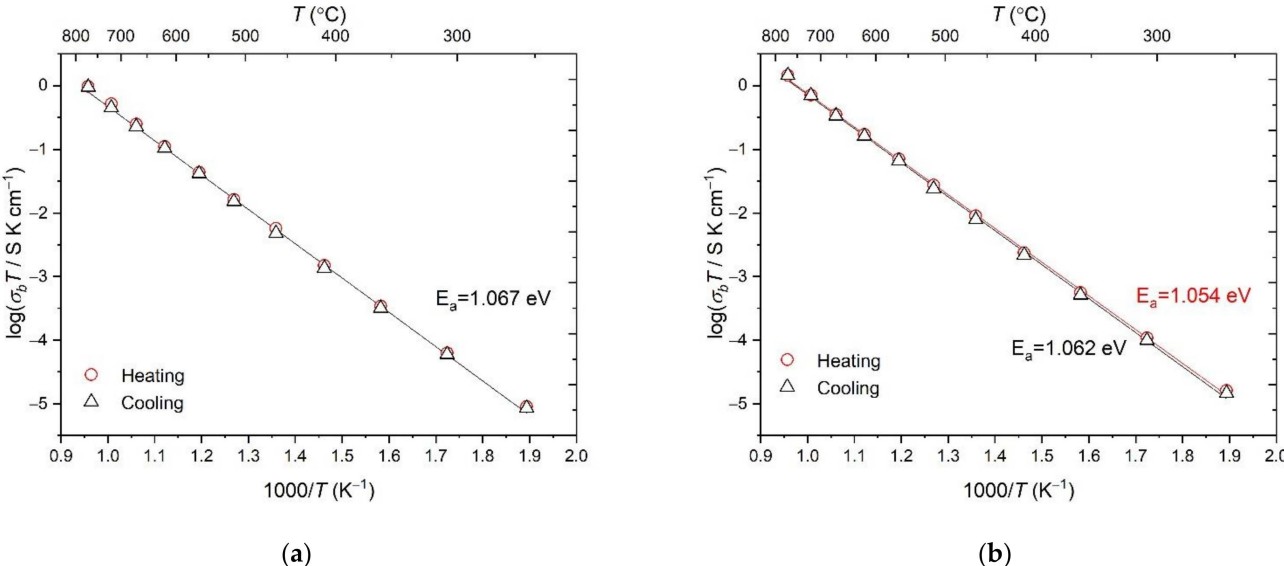

**Figure 8.** Arrhenius plots of bulk conductivity measured during heating and cooling for the (**a**) rapidly and (**b**) slowly cooled $Tm_2(Ti_{2-x}Tm_x)O_{7-x/2}$ ($x = 0.28$) solid solution.

Considering the change in the slope of the temperature dependence of the conductivity of the fluorite-like $Tm_2(Ti_{2-x}Tm_x)O_{7-x/2}$ ($x = 0.74$) at 550 °C (Figure 5a–c), the conductivity of a slowly cooled sample was measured not only in dry but also in wet air. Figure 9a,b show the Arrhenius dependence of the bulk and total conductivity for rapidly cooled (in dry air) and slowly cooled (in dry and wet air) $Tm_2(Ti_{2-x}Tm_x)O_{7-x/2}$ ($x = 0.74$) fluorite, respectively. It can be seen that decreasing the cooling rate in the range of 1600–1300 °C does not affect the temperature dependence of the bulk conductivity of fluorite, but it is obvious that the bulk conductivity is higher in wet air than in dry air (Figure 9a, Table 3). Taking into account the almost 100% density of this ceramic (Figure 3f), we can identify the proton contribution to the conductivity, which was first discovered in the $Ln_2O_3$-$TiO_2$ ($Ln = Dy - Lu$) systems. The proton transfer number has a value of about 0.7 (inset Figure 9a) in the temperature range from 255 °C to 411 °C and drops to 0.1 at 670 °C. The calculation is made according to the work [48].

Proton conductivity is only typical for fluorite $Tm_2(Ti_{2-x}Tm_x)O_{7-x/2}$ ($x = 0.74$). The low density of the $Tm_2(Ti_{2-x}Tm_x)O_{7-x/2}$ ($x = 0.67$) ceramics synthesized by solid state synthesis in [20] can explain the significant deviation of its temperature dependence of bulk conductivity at temperatures below 550 °C (Figure 5b). The temperature dependence of the total conductivity in dry air is slightly different for a rapidly and slowly cooled sample (Figure 9b), and this difference is only associated with a change in the grain boundary conductivity of ceramics (Figure 9a,b).

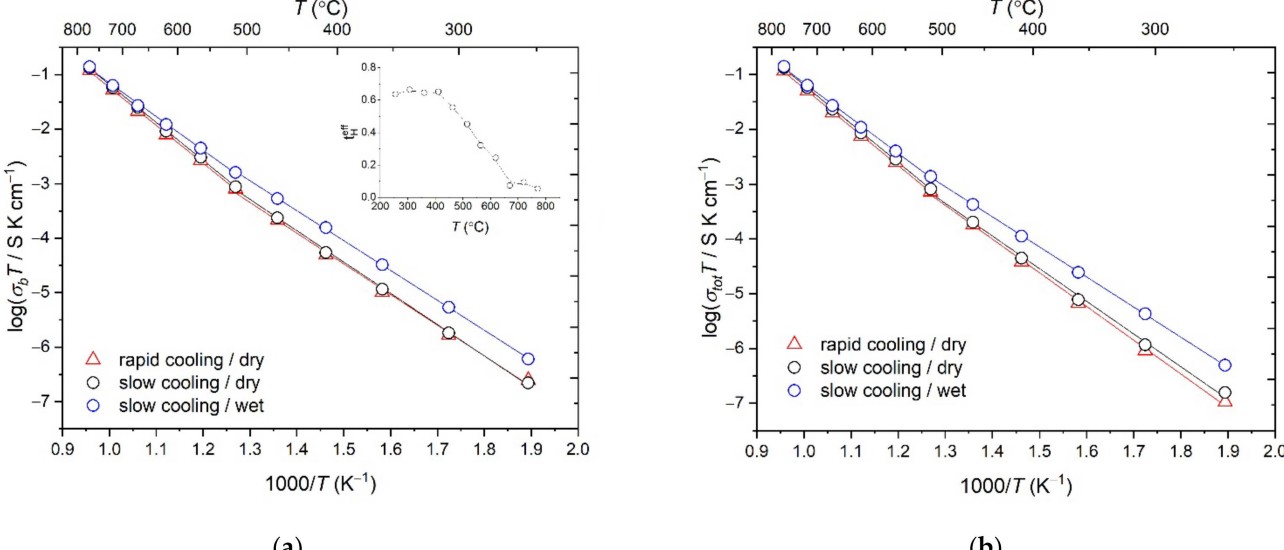

**Figure 9.** Arrhenius plots of (**a**) bulk and (**b**) total conductivity for rapidly cooled (in dry air (red triangles)) and slowly cooled (in dry (black circles) and wet air (blue circles)) fluorite $Tm_2(Ti_{2-x}Tm_x)O_{7-x/2}$ ($x = 0.74$).

*3.4. Dielectric Properties of the Solid Solutions*

Figure 10 shows temperature dependences of permittivity at low frequencies (0.5–250 Hz) for the $Tm_2(Ti_{2-x}Tm_x)O_{7-x/2}$ ($x = 0, 0.1, 0.18, 0.28, 0.74$) "stuffed" solid solutions. Peaks in temperature dependences of permittivity may have various origins. In our case, such a peak was observed at the lowest frequency, 0.5 Hz, in the curves of the nominally stoichiometric and slightly substituted pyrochlores. Such effects can be due to (ferroelectric) phase transitions [49–51] and growth transitions in materials [23]. They can reflect oxidation and reduction processes in ceramics in the temperature range under study [52] and can be related to defects in the materials. In our case, the permittivity peak at a frequency of 0.5 Hz (Figure 10a) is characteristic of the nominally stoichiometric pyrochlore phase. $Tm_2Ti_2O_7$ pyrochlore shows a "floating" maximum in permittivity, due to oxygen vacancy relaxation in the range 600–700 °C (Figure 10a). Here all oxygen vacancies participate in the hopping mechanism of oxygen–ion conductivity, typical of oxygen–ion conductors. A similar peak was previously observed for $Tm_2Ti_2O_7$ [23] and was related to a defect fluorite, the formation of which always precedes the crystallization of pyrochlore [53].

As the Tm content of the $Tm_2(Ti_{2-x}Tm_x)O_{7-x/2}$ ($x = 0, 0.1, 0.18, 0.28, 0.74$) "stuffed" pyrochlores increases (Figure 10b–d), the temperature behavior of 0.5-Hz permittivity gradually changes: the peak in the curve disappears, and $\varepsilon'$ decreases. At the same time, the curves of the $Tm_2(Ti_{2-x}Tm_x)O_{7-x/2}$ ($x = 0.18, 0.28$) samples (Figure 10c,d) exhibit some bimodality, which can be interpreted as evidence for coexistence of similar (pyrochlore and fluorite) structures in short-range order. Note that there is no bimodality in the temperature dependence of permittivity for the fluorite-related $Tm_2(Ti_{2-x}Tm_x)O_{7-x/2}$ ($x = 0.74$) ceramic (Figure 10e).

*3.5. Causes of the Difference in Conductivity between the $Tm_2(Ti_{2-x}Tm_x)O_{7-x/2}$ Ceramics Prepared via Coprecipitation and by Solid-State Reaction and Single Crystals with the Pyrochlore Structure*

The conductivity of mixed oxides depends on extended defects as well, which can form during synthesis, e.g., if nanophase precursors are used [53]. An important role in the synthesis of high-temperature ceramics is played by the decomposition temperature of the precursor. For example, in a previous study where $Tm_2Ti_2O_7$ ceramics were prepared at 1670 °C [23], the co-precipitated precursor was decomposed at two temperatures: (1) below the crystallization exotherm in the DSC curve, at 650 °C, and (2) above the

exothermic peak, at 740 °C. Note that, in the former case, the ceramic prepared subsequently at high temperature had a higher conductivity. In the case of a wrong choice of the precipitate decomposition temperature and duration [53], residual hydroxycarbonates in the co-precipitated mixture can lead to the formation of extended defects in the ceramic after high-temperature firing.

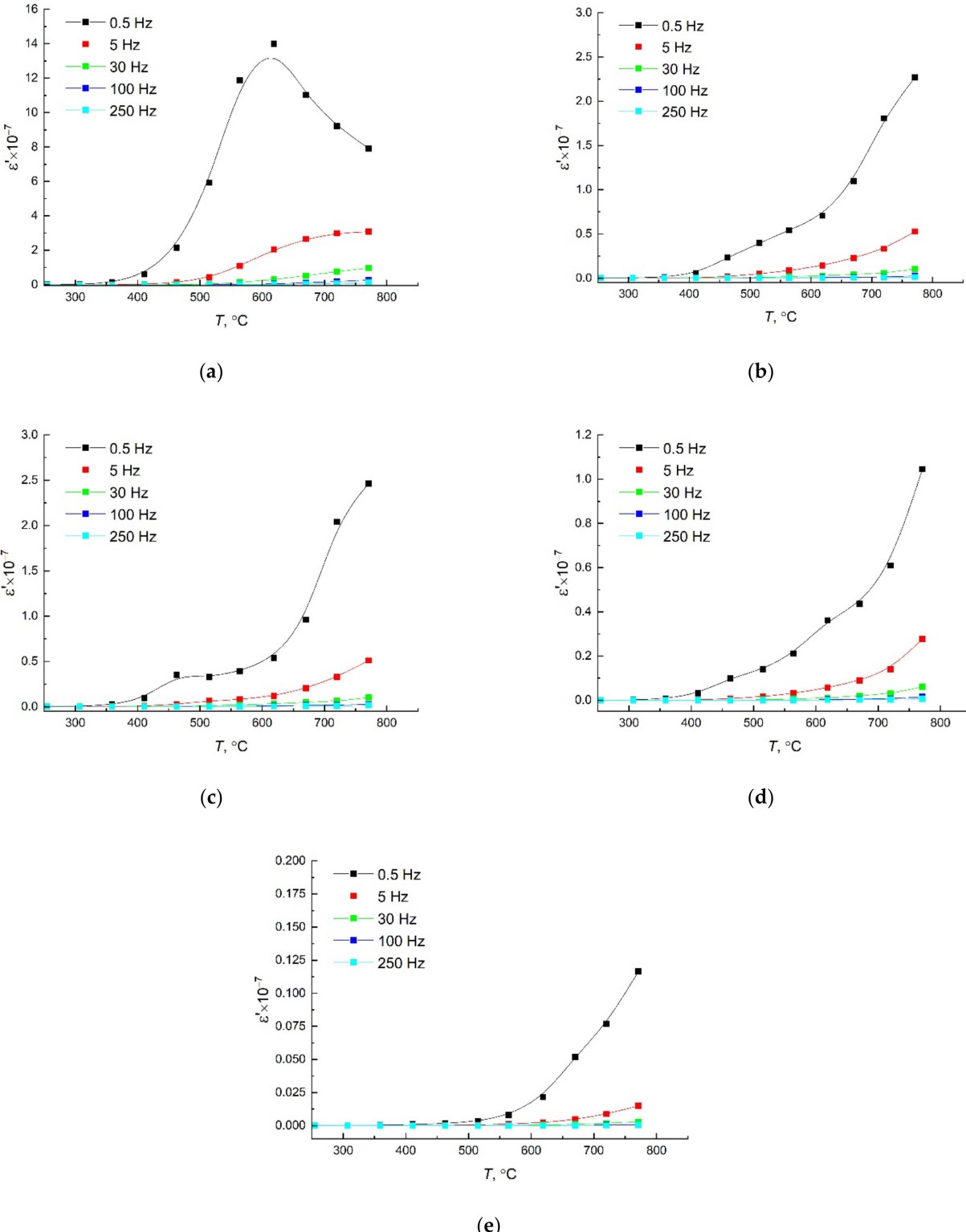

**Figure 10.** Temperature dependences of permittivity at different frequencies for the $Tm_2(Ti_{2-x}Tm_x)O_{7-x/2}$ "stuffed" solid solutions with $x$ = (**a**) 0, (**b**) 0.1, (**c**) 0.18, (**d**) 0.28, (**e**) 0.74.

Therefore, it is important to optimize nanophase precursor decomposition conditions in each particular case. For example, in the case of precursors prepared by hydroxide co-precipitation, the decomposition temperature should be ~100–150 °C below the crystallization peak of the pyrochlore phase in the DSC curve, and the decomposition time should be rather long [53]. In a number of recent studies [53–55], the mechanism of the synthesis of RE titanates and hafnates with the pyrochlore structure from nanophase precursors was clarified using thermal analysis, IR spectroscopy and Raman spectroscopy.

It is absolutely clear that, in syntheses via both co-precipitation and mechanical activation, reaction intermediates react with $CO_2$ and atmospheric moisture during milling or hydroxide co-precipitation. In connection with this, an important feature is the formation of basic RE carbonates ($Ln_2O_3 \bullet 2CO_2 \bullet 2H_2O$ and $LnOHCO_3 \bullet nH_2O$) and hydroxycarbonates ($Ln_2(CO_3)_2(OH)_{2(3-x)} \bullet n\,H_2O$) in intermediate synthesis steps [56,57]. Their decomposition in air is reversible up to 800–900 °C, so residual amounts of these compounds typically prevent one from obtaining dense ceramics [58–60].

If mixed oxides are prepared by solid-state reaction, which uses crystalline oxides, carbonates are even more difficult to decompose. Unlike solid-state reaction, synthesis methods utilizing nanophase precursors (co-precipitation, hydrothermal synthesis and mechanical activation in high-energy mills) produce a medium for active mass transport and allow one to minimize the amount of unreacted components. At the same time, thulium titanates prepared by different methods [20,23] differ in oxygen–ion conductivity by three orders of magnitude, so the above explanation is open to question.

In our opinion, studies of single crystals with the pyrochlore structure are important for understanding the origin of differences in conductivity between ceramics having identical or similar compositions. Single crystals were mainly grown for furthering the understanding of geometrically frustrated magnetism in $Ln_2Ti_2O_7$ ($Ln$ = Sm − Yb) [18,61–66]. The synthesis of large single crystals of many compounds in this series was carried out by the optical floating zone (OFZ) method [26,64]. $Yb_2Ti_2O_7$ single crystals grown by the same method but at different process parameters differed drastically in magnetic properties (specific heat anomaly). According to neutron diffraction results, one crystal had the stoichiometric composition (T = 1050 °C; slow growth speed (1.5 mm/h); air atmosphere) [65], whereas another $Yb_2Ti_2O_7$ single crystal (T = 1200 °C; slow growth speed (6 mm/h); 4 atm $O_2$) had a small deviation from stoichiometry: $Yb_2(Ti_{2-x}Yb_x)O_{7-x/2}$ ($x$ = 0.046) [26].

Recall that a flux-grown $Tm_2Ti_2O_7$ single crystal had a predominantly electronic conductivity at elevated temperatures [24], unlike the ceramics prepared in this work and having the same nominal composition. In addition to pyrochlores, $Gd_3Ga_5O_{12}$ garnet (GGG) single crystals can be referred to as "stuffed" because they were found to contain 1–2% excess Gd on the Ga sites. Gd accommodation on the Ga site occurs in a natural way during crystal growth [67].

Analysis of the above data for single crystals makes it possible to understand the origin of the considerable differences in oxygen–ion conductivity between ceramics having identical compositions but prepared by different methods. We believe that high firing temperatures ensure insignificant deviations from stoichiometry in materials synthesized from nanophase precursors. As a result, their composition approaches that of lightly doped "stuffed" pyrochlores, which typically have high oxygen–ion conductivity. It seems likely that the deviation from stoichiometry can be so small that it cannot always be detected by diffraction techniques. According to Ross et al. [26], possible XRD evidence of such a lightly doped lattice is an increased intensity of the pyrochlore 222, 440 and 622 lines. In this study, an increased intensity of these lines was observed in the case of the $Tm_2Ti_2O_7$ ceramic.

## 4. Conclusions

Precursors for the synthesis of "stuffed" $Tm_2(Ti_{2-x}Tm_x)O_{7-x/2}$ ($x$ = 0, 0.1, 0.18, 0.28, 0.74) pyrochlores have been prepared via co-precipitation, using aqueous ammonia as a precipitant. Optimization of precursor decomposition conditions has been shown to play an important role in the synthesis of dense ceramics at 1600 °C.

The highest oxygen–ion conductivity in the $Tm_2(Ti_{2-x}Tm_x)O_{7-x/2}$ ($x = 0$, 0.1, 0.18, 0.28, 0.74) "stuffed" pyrochlore series is offered by $Tm_2Ti_2O_7$. For the first time, proton conductivity has been found in fluorite $Tm_2(Ti_{2-x}Tm_x)O_{7-x/2}$ ($x = 0.74$).

Relaxation processes related to oxygen transport activity are characterized by a maximum in the temperature dependence of permittivity and are more pronounced at a low frequency of 0.5 Hz for the $Tm_2(Ti_{2-x}Tm_x)O_{7-x/2}$ ($x = 0$, 0.1) "stuffed" pyrochlores.

We have discussed the cause of the three orders of magnitude difference in conductivity between $Tm_2(Ti_{2-x}Tm_x)O_{7-x/2}$ ceramics prepared using co-precipitation and solid-state reaction and assumed that the use of high firing temperatures (1600–1670 °C) and nanophase precursors leads to the formation of lightly doped "stuffed" pyrochlores having high oxygen–ion conductivity.

Slow cooling initiates growth of fluorite nanodomains in a pyrochlore matrix and favors the formation of nanostructured dense "composites," which have great potential for use in SOFCs.

**Author Contributions:** A.S.: conceptualization, methodology, writing—original draft preparation, review and editing; N.G., E.B. and V.R.: investigation, formal analysis; N.G., E.B., O.K. and D.S.: investigation, data curation, visualization; N.G. and D.S.: resources; N.G. and E.B.: formal analysis, writing—review and editing. All authors have read and agreed to the published version of the manuscript.

**Funding:** The work was supported partially by the subsidy from the Ministry of Education and Science allocated by the FRC CP RAS for the implementation of the state assignment (No.122040500071-0) and in accordance with the state task for FRC PCP and MC RAS, state registration No. AAAA-A19-119061890019-5.

**Institutional Review Board Statement:** Not applicable.

**Informed Consent Statement:** Not applicable.

**Data Availability Statement:** Not applicable.

**Conflicts of Interest:** The authors declare no conflict of interest.

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
