# Peer review of "Oxygen–Ion Conductivity, Dielectric Properties and Spectroscopic Characterization of “Stuffed” Tm2(Ti2−xTmx)O7−x/2 (x = 0, 0.1, 0.18, 0.28, 0.74) Pyrochlores"

_ceramics, doi:10.3390/ceramics6020056_

Round 1

Reviewer 1 Report

The manuscript describes the synthesis, structural characterization and electrical properties of pyrochlores Tm2(Ti2−xTmx)O7−x/2. The object of article is very interesting. Authors use complex of investigation methods such as XRD, SEM, EIS. I think the results are interesting and they deserve to be published in Ceramics after minor revision.

1. The authors can improve the figure quality.

2. Please compare conductivity values of obtained compositions with most-known materials used as electrolytes in SOFCs.

Author Response

Dear reviewer,

Thank you for your remarks. We have changed the manuscript according to your recommendations.

Reviewer 1

Comments and Suggestions for Authors

The manuscript describes the synthesis, structural characterization and electrical properties of pyrochlores Tm2(Ti2−xTmx)O7−x/2. The object of article is very interesting. Authors use complex of investigation methods such as XRD, SEM, EIS. I think the results are interesting and they deserve to be published in Ceramics after minor revision.

  1. The authors can improve the figure quality.

Reply:

We have improved the figure quality.

We have changed new Figure 5b. We have changed the legend for clarity:

Tm2(Ti1.23Tm0.67)O6.47- 50% Tm2O3[20].

  1. Please compare conductivity values of obtained compositions with most-known materials used as electrolytes in SOFCs.

Reply:

Thank you! We have added this information to the text.

Recently, a maximum oxygen-ion conductivity of ~9.4×10-2 S/cm at 800º C has been achieved on 5 mol% Bi2O3 doped ScSZ (Sc2O3 doped ZrO2) [37]. However, the introduction of Bi2O3 oxide has its negative side associated with its easy reduction.

[37] H. Wang, Z. Lei, W. Jiang, X. Xu, J. Jing, Z. Zheng , Z. Yang, S. Peng.  Intern. High-conductivity electrolyte with a low sintering temperature for solid oxide fuel cells. Intern. J. Hydr. Energy 2022, 47, 11279–11287. https://doi.org/10.1016/j.ijhydene.2022.01.168

Reviewer 2 Report

In this work entitled “Oxygen-Ion Conductivity, Dielectric Properties and Spectroscopic characterization of “Stuffed” Tm2(Ti2−xTmx)O7−x/2 (x = 0, 0.1, 0.18, 0.28, 0.74) Pyrochlores”, Tm2(Ti2−xTmx)O7−x/2 (x = 0, 0.1, 0.18, 0.28 and 0.74) oxides have been synthesized and evaluated as potential electrolytes for solid oxide fuel cells. The crystal structure properties of the Tm2(Ti2−xTmx)O7−x/2 solid solutions were characterized by the XRD and Optical Spectroscopy studies. The morphology and microstructure of the Tm2(Ti2−xTmx)O7−x/2 (x = 0, 0.1, 0.18, 0.28, 0.74) ceramics were investigated by the SEM analysis. The electrical conductivity data of  Tm2(Ti2−xTmx)O7−x/2 (x = 0, 0.1, 0.18, 0.28, 0.74) were collected by the Impedance Spectroscopy technique. In addition, the dielectric properties of the Tm2(Ti2−xTmx)O7−x/2 solid solutions were also characterized. Interestingly, Tm2(Ti2−xTmx)O7−x/2 (x = 0.74) fluorite presents protonic conductivity of ~ 5×10−5 S/cm at 600 °C.

In general, the work is of interest for the readers of the journal of Ceramics, and the following suggestions could be considered by the authors for a further improvement of the manuscript.  

1.      Please remove the statement of “stack” and “SOFC batteries”, which is rather confusing for the readers. The solid electrolyte is the basic element for SOFC. All abbreviations should be defined at first mention in the manuscript.

2.      The authors stated in line 14 - “potentially operating in the medium temperature range (600–700 °C) using methane as fuel”, however, in the manuscript there is no work/evidence to support the cell fueled by methane.

3.      The choice of Tm2(Ti2−xTmx)O7−x/2 (x = 0, 0.1, 0.18, 0.28 and 0.74) composition should be justified. Why such as x=0.5 or 0.6 not prepared.

4.      In Figure 1, the residual of the fitting is large. The authors may also refine the Uiso value.

5.      The proton transfer number could be possibly calculated from the total and proton conductivity data.

6.      The chapter of 3.3 and 3.4 should be combined. Figure 5a and Figure 6a can be merged.

7.      The manuscript should be carefully rechecked, and editing of English language and style is required throughout the manuscript due to too mistakes.

Author Response

Dear reviewer,

Thank you for your remarks. We have changed the manuscript according to your recommendations.

Reviewer 2

Comments and Suggestions for Authors

In this work entitled “Oxygen-Ion Conductivity, Dielectric Properties and Spectroscopic characterization of “Stuffed” Tm2(Ti2−xTmx)O7−x/2 (x = 0, 0.1, 0.18, 0.28, 0.74) Pyrochlores”, Tm2(Ti2−xTmx)O7−x/2 (x = 0, 0.1, 0.18, 0.28 and 0.74) oxides have been synthesized and evaluated as potential electrolytes for solid oxide fuel cells. The crystal structure properties of the Tm2(Ti2−xTmx)O7−x/2 solid solutions were characterized by the XRD and Optical Spectroscopy studies. The morphology and microstructure of the Tm2(Ti2−xTmx)O7−x/2 (x = 0, 0.1, 0.18, 0.28, 0.74) ceramics were investigated by the SEM analysis. The electrical conductivity data of  Tm2(Ti2−xTmx)O7−x/2 (x = 0, 0.1, 0.18, 0.28, 0.74) were collected by the Impedance Spectroscopy technique. In addition, the dielectric properties of the Tm2(Ti2−xTmx)O7−x/2 solid solutions were also characterized. Interestingly, Tm2(Ti2−xTmx)O7−x/2 (x = 0.74) fluorite presents protonic conductivity of ~ 5×10−5 S/cm at 600 °C.

In general, the work is of interest for the readers of the journal of Ceramics, and the following suggestions could be considered by the authors for a further improvement of the manuscript.  

  1. Please remove the statement of “stack” and “SOFC batteries”, which is rather confusing for the readers. The solid electrolyte is the basic element for SOFC. All abbreviations should be defined at first mention in the manuscript.

Reply:

We have changed the text.

  1. The authors stated in line 14 - “potentially operating in the medium temperature range (600–700 °C) using methane as fuel”, however, in the manuscript there is no work/evidence to support the cell fueled by methane.

Reply:

We have changed the text.

  1. The choice of Tm2(Ti2−xTmx)O7−x/2 (x = 0, 0.1, 0.18, 0.28 and 0.74) composition should be justified. Why such as x=0.5 or 0.6 not prepared.

Reply:

In a similar series for LuTiO ceramics, a decrease in oxygen-ion conductivity was observed for the composition x ~ 0.5 [1].

[1] Shlyakhtina, A.V.; Abrantes, J.C.C.; Levchenko, A.V.; Knotko, A.V.; Karyagina, O.K.; Shcherbakova, L.G. Synthesis and electrical transport of Lu2+xTi2-xO7-x/2 oxide-ion conductors. Solid State Ionics (2006),177, 1149-1155.

  1. In Figure 1, the residual of the fitting is large. The authors may also refine the Uiso value.

          Reply: We have recalculated the data and changed Figure 1.

  1. The proton transfer number could be possibly calculated from the total and proton conductivity data.

Reply:

We have presented the proton transport number (inset Figure 9a). The calculation is made according to the work [49].

The proton transfer number has a value of about 0.7 (inset Figure 9 a) in the temperature range from 255℃ to 411℃ and drops to 0.1 at 670℃.

[49] Putilov L.P.; Tsidilkovski V.I. Improving the performance of protonic ceramic fuel cells and electrolyzers: The role of acceptor impurities in oxide membranes. Energy Convers. Manag., 2022, 267, 115826.

  1. The chapter of 3.3 and 3.4 should be combined. Figure 5a and Figure 6a can be merged.

Reply:

We have done.

  1. The manuscript should be carefully rechecked, and editing of English language and style is required throughout the manuscript due to too mistakes.

Reply: We have tried to improve the English language.
